# Non-Severe Hypophosphatemia in Older Patients: A Systematic Review

**DOI:** 10.3390/nu17081354

**Published:** 2025-04-16

**Authors:** Luca Barbarossa, Martina Zandonà, Maria Luisa Garo, Ribal Bou Mjahed, Patrizia D’Amelio

**Affiliations:** 1Geriatric Service, Clinica Luganese di Moncucco, 6900 Lugano, Switzerland; 2Family Medicine Institute, Ente Ospedaliero Cantonale, 6900 Lugano, Switzerland; 3Biostatistic Unit, Mathsly Research, 00128 Rome, Italy; marilu.garo@mathsly.it; 4Service of Geriatric Medicine and Geriatric Rehabilitation, Centre Hospitalier Universitaire Vaudois (CHUV), 1005 Lausanne, Switzerland; ribalboumjahed@gmail.com (R.B.M.); patrizia.damelio@chuv.ch (P.D.)

**Keywords:** phosphorus, hypophosphatemia, older adults, geriatrics, frailty, unhealthy aging

## Abstract

**Background/Objectives:** Phosphorus plays a fundamental role in cellular and extracellular metabolism, contributing to nucleic acid synthesis, enzymatic activity, neurologic function, and skeletal mineralization. Despite its significance, non-severe hypophosphatemia (HP) remains largely asymptomatic and underdiagnosed, with limited data on its prevalence in the general population. Most studies focus on specific subgroups, such as critically ill or dialysis patients, while the impact of mild HP in older adults, a potentially vulnerable demographic, is not well understood. The objective of this systematic review is to investigate the prevalence and clinical implications of non-severe HP in older adults. **Methods:** The study followed PRISMA guidelines to assess HP in patients aged ≥ 65 years without critical illnesses or genetic disorders. A systematic search was conducted in PubMed, Web of Science, and Scopus (March 2024). Eligible studies included RCTs and prospective/retrospective studies, excluding cancer-related studies or insufficient phosphate data. **Results:** We identified 1350 articles, with 26 meeting eligibility criteria. Ultimately, eight studies involving 26,548 patients were included, with an HP prevalence of 12.5%. Studies reveal a higher prevalence of HP in frail individuals, particularly those with increased frailty scores, and an association between HP and cognitive decline, depressed mood, falls, and chronic comorbidities. HP was also prevalent in infectious diseases, especially bacterial pneumonia, with longer hospital stays and increased mortality rates. Malnutrition was significantly more common in HP patients, characterized by weight loss and poor nutritional status. Furthermore, HP increased fall risk during hospitalization and worsened outcomes after coronary artery bypass graft surgery, including higher mortality and graft failure rates, underscoring its clinical importance. **Discussion:** This review identified that non-severe hypophosphatemia (HP) is associated with conditions such as frailty, cognitive decline, and an increased risk of falls. The evidence suggests that low phosphate levels may negatively impact health, increasing mortality and the risk of adverse clinical outcomes. Despite limitations in diagnostic criteria and sample variability, the findings indicate that HP can be a useful marker for identifying patients at risk of health deterioration. Further research is needed to clearly define the management and treatment of HP in this vulnerable population.

## 1. Introduction

Phosphorus plays an essential biochemical role as it is involved in both cellular and extracellular metabolism [1,2]. It is indeed a component of nucleic acids, cell membranes, enzyme systems, and numerous nucleoproteins, and it is also involved in neurologic functions and skeletal mineralization [3,4]. Phosphorus is an important component of the hydroxyapatite crystals present in bones and teeth, where it is stored bound with calcium. The skeleton represents the main phosphorus storage in the human body (about 85%) [5], while only 14% is present as phosphate in soft tissues and the remaining 1% as inorganic phosphate in the extracellular fluids. Although serum inorganic phosphate represents a very small percentage of total phosphorus in the body, its measurement can provide useful information about the status of body storage. The regulation of phosphorus metabolism is complex, controlled by several hormones, influenced by diet, and modulated by plasma pH. The main regulators of phosphorus homeostasis are Fibroblast Growth Factor 23 (FGF23), Parathyroid Hormone (PTH), and 1,25-dihydroxyvitamin D (1,25[OH]2D) [6,7].

Normal phosphate serum levels in adults range between 2.5 and 4.5 mg/dL (0.81 to 1.45 mmol/L). Non-severe hypophosphatemia (HP) is defined by a serum phosphate level lower than 2.5 mg/dL (0.81 mmol/L), while severe HP is identified with a threshold of less than 1 mg/dL (0.32 mmol/L). The prevalence of non-severe HP in the general population is unknown, as it is usually asymptomatic and phosphorus is not routinely measured [8]. The prevalence of HP in hospitalized patients ranges from 2.2% to 3.1%; these percentages rise to 28.8% to 34% in patients hospitalized in intensive care units. Hypophosphatemia is diagnosed in 2.5% to 30.4% of patients with chronic alcohol consumption, in 65% to 80% of patients with severe sepsis, and in up to 75% of major trauma patients [9]. In general practice patients, HP has been found in 8–12% regardless of their age [10].

Four main mechanisms are responsible for HP: the redistribution of phosphate from extracellular fluid into intracellular space or bone, decreased intake or intestinal absorption, increased urinary excretion, and removal by dialysis [11]. The shift of phosphate from extracellular to intracellular space may be caused by increased insulin secretion, especially during the refeeding syndrome [12] or diabetic ketoacidosis [13], acute alkalosis [14], and hungry bone syndrome after parathyroidectomy [15]. Reduced intake is a rare cause of HP as phosphate is ubiquitous and the kidney is able to compensate for decreased intestinal absorption by increasing phosphate absorption [16]. It is more common in cases of chronic alcohol abuse [17] or phosphate malabsorption due to steatorrhea and chronic diarrhea or the use of medications that can inhibit intestinal absorption, such as aluminum- and magnesium-based antacids [18]. Hypophosphatemia is usually due to increased urinary excretion, which can be caused by primary hyperparathyroidism, intrinsic renal defects such as Fanconi syndrome, and increased FGF23 secretion [19]. FGF23 is increased in genetic conditions such as X-linked hypophosphatemic rickets [20], tumor-induced osteomalacia, and intravenous iron administration [21]. Increased urinary phosphate excretion is also a side effect of several classes of chemotherapeutic agents (Imatinib, Temsirolimus, VEGF-inhibitors, and others) [22,23,24] and antiretroviral drugs such as Tenofovir [25]. Finally, HP is observed in many patients undergoing dialysis [26].

While severe HP can lead to anorexia, muscle weakness, osteomalacia, neuromuscular disorders, seizures, coma, and death, non-severe HP is generally asymptomatic [27]. Being a frequently incidental finding, mild HP prevalence and related clinical outcomes are currently unclear.

Most studies on HP focused on specific subgroups of patients, such as oncological patients, patients on hemodialysis, patients with genetic conditions causing low serum phosphate levels, critically ill patients, and refeeding syndrome but not on the general population of older adults.

Thus, the aim of this systematic review is to explore the prevalence and possible clinical effects of non-severe HP in older adults without specific diseases associated with severe forms of HP.

## 2. Materials and Methods

This work was conducted in accordance with the PRISMA guidelines [28]. The components of the PECO question were as follows. (Population): patients ≥ 65 years without critical illnesses or genetic diseases causing HP or hemodialysis patients; (Exposure): patients with HP (Comparator): patients without HP; and (Outcomes): potentially negative clinical outcomes, comorbidities, or mortality associated with HP. The review protocol is published on PROSPERO database (PROSPERO CRD42021252144.).

### 2.1. Eligibility Criteria

Peer-reviewed research articles written in English were considered for inclusion; the studies included were randomized controlled trials (RCTs) and prospective or retrospective studies; literature reviews, clinical cases, case reports, letters to the editor, and conference abstracts and animal studies were excluded. To properly study the effects of non-severe HP in older adults under typical clinical conditions and minimize confounding factors, we included studies involving subjects aged 65 years or older. Included studies have to assess non-severe HP, exploring its clinical association (e.g., frailty, cognitive impairment, malnutrition, falls, and risk of infection) and explicitly reported phosphate levels along with clear information on non-severe HP and clinical outcomes. Studies focusing on advanced oncologic disease, critically ill patients or patients in intensive care, individuals undergoing hemodialysis, or individuals affected by rare genetic syndromes were excluded, as these conditions are known to affect phosphate metabolism. Additionally, studies were excluded if they lack a clear hypophosphatemia definition or sufficient clinical data.

### 2.2. Search Strategy

A systematic search strategy was performed in PubMed, Web of Science, and Scopus during March 2024 (last research run on March 20) without time or language restriction using the following keywords: (((hypophosphatemia[MeSH Terms]) OR (phosphorus supplementation)) AND ((elderly[MeSH Terms]) OR (older adult*))) NOT ((critical illness[MeSH Terms]) OR (care intensive unit OR ICU[MeSH Terms])). The search strategy is available in the Appendix A.

The removal of duplicates and screening of titles/abstracts was performed by two independent reviewers (MZ and MLG). The full texts of the remaining potentially relevant articles that met the inclusion and exclusion criteria were retrieved and reviewed by two independent reviewers (MZ and LB). Any discrepancies were discussed until a consensus decision was reached. The final eligibility of each study was reviewed and the reasons for exclusion were recorded. Three authors (MZ, MLG, and LB) made the final selection of articles. Disagreements were discussed between all authors. A hand-search for suitable reports was performed through citation screening of included studies.

### 2.3. Data Extraction

Two authors (MZ and MLG) independently extracted data from the full texts of studies that met the inclusion criteria. Discrepancies were resolved by team discussion.

The data collected included

Study characteristics: first author, publication year, country, observation period, study design;Setting: hospital vs. outpatient department;Inclusion criteria;Sample size;Patient’s mean or median age;Patients’ sex;No. of patients with HP;HP cut-off at diagnosis;Phosphorus mean/median value;Comorbidities;Primary and secondary outcomes.

### 2.4. Risk of Bias-Quality Assessment

The quality of the included studies was assessed independently by three reviewers (MZ, MLG, and LB) using the Quality Assessment Tool for Observational Cohort (https://www.nhlbi.nih.gov/health-topics/study-quality-assessment-tools, accessed on 1 February 2024). The tool consists of 14 items aimed at assessing the clear presentation of the study’s research questions or objectives, the definition of the population and the proper explanation of inclusion/exclusion criteria, statistical aspects (such as justification of sample size and adjusted analysis for confounding variables), and proper assessment of exposure and outcomes. Potential disagreements were resolved by discussion and consensus among all authors.

## 3. Results

### 3.1. Search Results

One thousand seven hundred and twenty-five papers were identified (PubMed: 613, Scopus: 892, Web of Science: 220). After removing duplicates (*n* = 375), 1350 articles were screened by title and abstract. After excluding 1323 articles, 26 articles were eligible for full-text analysis; 18 articles were excluded (Table 1) because the sample mean age was below 65 years (*n* = 9 studies), HP status was not reported (*n* = 7 studies), and the total sample mainly consisted of cancer patients (*n* = 2). A complete overview of the study selection is shown in Figure 1.

### 3.2. Studies Characteristics

Eight studies, mainly retrospective studies, with a total of 26,548 patients, of whom 3321 suffered from HP (prevalence of 12.5%), were included in this paper. The studies were conducted between 1997 and 2022. Two studies were conducted in Japan (Fujisawa et al. [30] and Morimoto et al. [31]), two in South Korea (Jang et al. [32] and Park et al. [33]), one in Sweden (Haglin et al. [34]), one in Turkey (Heybeli et al. [35]), and one in Germany (Pourhassan et al. [36]) and one in USA (Sankaran et al. [37]). Six of the eight studies included a full description of the demographic characteristics of the patients, while in two studies, the sex of the patients was not specified [36,37]. The definition of HP was not consistent across the studies: there were several cut-off values ranging from 2.0 mg/dL to 3.0 mg/dL. The role of HP was investigated in relation to certain diseases/comorbidities: Fujisawa et al. (2022) [30] and Heybeli et al. (2022) [35] investigated a possible effect of HP related to electrolyte imbalance, Haglin et al. (2010) [34] calculated the prevalence of HP in older patients with influenza, Jang et al. (2022) [32] determined the effect of HP on the risk of falls during hospitalization, Sankaran et al. (1997) [37] and Morimoto et al. (2022) [31] evaluated a possible role of HP on the prognosis of patients with pneumonia, Park et al. (2019) [33] investigated the role of HP on the outcome of coronary artery bypass grafting, and Pourhassan et al. (2018) [36] investigated the association between HP and malnutrition.

The study characteristics and patients’ enrolment criteria are listed in Table 1a and Table 1b, respectively.

**Table 1 nutrients-17-01354-t001:** (**a**) Studies characteristics. (**b**) Patients’ enrolment criteria.

(a)							
Authors	Country	Period	Design	Setting	Sample	S-P Level (mg/dL)	HP pz.
Fujisawa et al. (2022) [30]	Japan	2010–2017	Cross-sectional study	Outpatient department	4204	3.0	370
Haglin et al. (2010) [34]	Sweden	1992–1994	Retrospective	Hospital	76 *	2.5	24
Heybeli et al. (2022) [35]	Turkey	2016–2020	NR	Outpatient department	464	2.5	23
Jang et al. (2022) [32]	South Korea	2018–2021	Retrospective	Hospital	15,485	2.8	2406
Morimoto et al. (2022) [31]	Japan	2009–2018	Retrospective	Hospital	600	2.0	72
Park et al. (2019) [33]	South Korea	2010–2014	Retrospective	Hospital	4782	2.5	238
Pourhassan et al. (2018) [36]	Germany	NR	Retrospective	Hospital	NR	335	2.1
Sankaran et al. (1997) [37]	USA	1993	Retrospective	Hospital	NR	602	2.4
(**b**)							
**Authors**	**Inclusion Criteria**		**Age**		**Female (%)**	**Comorbidities (%)**	
Fujisawa et al. (2022) [30]	Patients aged 70 years or older visited forassessment of a memory disorder who present at least one among serumNa, K, Ca, P and answered for at least 30 items on the 50-item FI.		Non-frail group: 75 (73–79); Mildly frail group: 78 (75–82); Moderate frail group: 80 (77–84); Severe frail group: 83 (79–87)		563 (61.1) in the frail group 691 (60.3) in the mildly frail group 698 (67.2) in the moderately frail group 764 (69.5) in the severely frail group	Diabetes 1500 (35.7); Hypertension 2129 (50.6); Heart disease 531 (12.6); Liver disease 101 (2.4); Lung disease 177 (4.2); Cancer 30 (7.8); Stroke 253 (6.0); Estimated glomerular filtration rate <60 mL/min/1.73 m^2^: 1504 (35.8); Insomnia complaint 154 (3.7)	
Haglin et al. (2010) [34]	Patients with virologically confirmed acute influenza		Unclear		25 (50)	Diabetes: 13 (17.1); Bronchial asthma: 4 (5.3); CVD: 4 (5.3); COPD: 4 (5.3); Other: 10 (13.2)	
Heybeli et al. (2022) [35]	Patients aged ≥ 65 years		78 (72–83)		321 (69.2)	Diabetes: 167 (36); Hypertension: 320 (69); Chronic kidney disease: 181 (39); Heart failure: 51 (11); Ischemic heart disease: 70 (15); Cerebrovascular disease: 60 (13)	
Jang et al. (2022) [32]	Hospitalized patients		70.0 (60.0–79.0)		7971 (52.1)	NR	
Morimoto et al. (2022) [31]	Hospitalized patients with community-acquired pneumonia		67.9 ± 15.2		175 (29.2)	Pulmonary disease: 321 (53.5); Non-pulmonary disease: 312 (52)	
Park et al. (2019) [33]	Patients undergoing CABG		Normal: 63.1 ± 9.8 HP: 65.1 ± 9.6		1237 (25.9)	Hypertension: 2957 (61.8) ; Diabetes: 2212 (46.3) ; Ejection fraction < 40%: 1398 (29.2) ; Dyslipidemia: 1598 (33.4) ; Stroke: 721(15.1); Chronic kidney disease: 243 (5.1); COPD: 1479 (30.9); PAOD: 360 (7.5) ; ACS: 2387 (49.9); Old MI: 587 (12.3); Carotid arterial disease: 1082 (22.6)	
Pourhassan et al. (2018) [36]	Older hospitalized patients		83.1 ± 6.8		NR	NR	
Sankaran et al. (1997) [37]	Hospitalized patients		HP: 67.5 ± 1. 9; Control group: 60.7 ± 1.1 years; Normophosphatemia: 63.4± 1.5 years		NR	NR	

* The prevalence of HP was determined on 50 patients. NR: Not Reported.

A graphical representation was created to summarize and visually depict the main clinical associations of non-severe HP reported in the included studies. The nodes in the diagram (Figure 2) represent either the central exposure (non-severe HP) or associated conditions, which were categorized into clinical, infectious, functional, surgical, and prognostic domains. The number of studies explaining this association is indicated in each diagram.

### 3.3. Risk of Bias

The risk of bias analysis revealed significant shortcomings in the conduct of the included studies. The main concerns were the lack of a clear definition of the research question [35,37], an inadequate definition of the sample population [35], and an inadequate definition of the outcomes [35]. The statistical analyses of four studies appeared to be affected by bias due to unspecified sample size [31,35,36,37]. In three studies, potential confounding variables were not assessed; consequently, no statistical adjustment was made [34,35,36]. The full assessment of the risk of bias is presented in Appendix A (Appendix A) and Figure 3 and Figure 4.

### 3.4. Hypophosphatemia and Related Comorbidities

Two studies investigated the possible role of HP in the context of in-depth analyses related to electrolyte imbalances (Table 2). Fujisawa et al. (2022) [30] showed, in a cross-sectional study conducted on a sample of 4204 patients aged 70 years or older, that mildly (0.20 < Frailty Index score ≤ 0.3), moderately (0.3 < Frailty Index score < 0.4) and severely (Frailty Index score ≥ 0.4) frail patients were more likely to suffer from HP compared to non-frail patients (Frailty Index score ≤ 0.2) (mildly frail group: OR, 1.52; 95% CI, 1.07–2.14; *p* = 0.02; moderately frail group: OR, 1.56; 95% CI, 1.08–2.24; *p* = 0.02; severely frail group: OR, 2.00; 95% CI, 1.37–2.89; *p* < 0.001) and that each 0.1 increase in Frailty Index score led to a 16% increase in HP risk (exp(β) = 1.16, 95%CI: 1.06–1.27). They also demonstrated that HP was associated with an increased risk of cognitive impairment and depressed mood (OR: 1.07, 95%CI: 1.03–10.12, *p* < 0.01), an increased risk of falling due to physical weakness (OR: 1.05, 95%CI: 1.01–1.10, *p* < 0.05), and comorbidities (OR: 1.05, 95% CI: 1.00–1.10, *p* < 0.05).

In the same year, Heybeli et al. (2022) [35] conducted a study aimed at investigating abnormal sodium, potassium, calcium, phosphorus, and magnesium levels and their association with functional dependency in a sample of older patients admitted to a single geriatric outpatient clinic. The authors defined HP as phosphorus levels below 2.4 mg/dL and diagnosed HP in 5% out of 464 patients (median phosphate: 2.3 (2.1–2.4) mg/dL); they observed lower prevalence of diabetes mellitus in patients with HP (13.6%) compared to those without HP (37%, *p* = 0.026). Although not statistically significant, the prevalence of HP was higher in patients aged over 80 years (6.7%) than in those aged 65–79 years (3.7%), as well as in patients with chronic kidney disease. No statistically significant association was found between non-severe HP and functional dependency. The latter was measured by the Barthel Activities of Daily Living and the Lawton–Brody Instrumental Activities of Daily Living. Additionally, a quarter of HP patients were treated with polypharmacy (defined as treatment with more than three different drugs), compared to only 3.3% of patients without HP (*p* = 0.057).

### 3.5. Infectious or Bacterial Diseases

In three studies (a total of 1278 patients; prevalence of HP: 20.5%), a possible connection between infections and low phosphorus levels was investigated (Table 2). In 2010, Haglin et al. [34] suggested a possible role of HP in a sample of 50 patients hospitalized for influenza and recorded a prevalence of HP (S-P ≤ 0.82 mmol/L) in 34% of patients with a higher prevalence of mild–severe HP (S-P < 0.70 mmol/L) in 13% of women and 15% of men. The simultaneous presence of influenza and a chronic disease (e.g., diabetes mellitus, bronchial asthma, cardiovascular disease, or chronic obstructive pulmonary disease) increased the risk of HP almost two-fold.

In a study by Morimoto et al. [31], which was conducted retrospectively over a 10-year period on a sample of 600 patients hospitalized for bacterial pneumonia, HP (<2.0 mg/dL) was found in 72 patients (12%), 19 of whom had severe pneumonia. The prevalence of non-pulmonary disease and the proportion of severe pneumonia was significantly higher in patients with HP (65.3%) than in patients without HP (50.2%, *p* = 0.017). Pathogens causing pneumonia were equally common in patients with and without HP (HP: 54.2%, non-HP: 50.2%), while polymicrobial infections were more common in HP patients (*n* = 9, 12.5%) than in non-HP patients (*n* = 44, 8.3%). Legionella spp. was significantly more frequently identified as an etiology in HP patients than in non-HP patients (11.1% vs. 3.6%, *p* = 0.010). Although not significant, a higher percentage of HP patients with viruses other than influenza had HP (6.9%). In a multivariate analysis, the authors showed that Legionella spp., diabetes mellitus, and severe pneumonia were independent factors for HP (Legionella spp. OR: 2.89; 95% CI, 1.19 to 6.99; *p* = 0.019, diabetes mellitus OR: 2.53; 95% CI, 1.41 to 4.56; *p* = 0.002, severe pneumonia OR: 2.86; 95% CI, 1.57 to 5.22; *p* = 0.001).

In a previous study conducted on 302 patients with pneumonia and 300 controls, Sankaran et al. (1997) [37] showed that HP was more common in patients diagnosed with bacterial pneumonia at hospital discharge (44.7%) than in patients who did not have bacterial pneumonia at hospital discharge (10.3%, *p* < 0.001). HP patients had significantly lower potassium (HP: 4.2 ± 01 mmol/L), calcium (HP: 8.3 ± 0.1 mg/dL), and albumin (HP: 2.7 ± 0.1 g/dL) levels than non-HP patients (potassium: 4.4 ± 0.1 mmol/L, *p* < 0.05, calcium: 8.6 ± 0.1 mg/dL, *p* < 0.01, and albumin: 3.0 ± 0.1 g/dL, *p* < 0.01) but, unlike the non-HP patients, had higher glucose levels (HP: 151 ± 6 mg/dL, non-HP: 129 ± 5 mg/dL, *p* < 0.01). Hospital stays were significantly longer in HP patients (HP: 24.6 ± 2.1 days versus non-HP, 14.1 ± 1.0; *p* < 0.0001) and mortality rate was higher in HP patients (≃30%) than in non-HP patients (≃14%, *p* < 0.001).

### 3.6. Risk of Falls

In a Korean study conducted in 2022 on a sample of hospitalized patients with various diagnoses, Jang et al. [32] found that the incidence of falls during hospitalization was significantly higher in patients with lower serum phosphate levels (about 2.3%), which was twice as high as in patients with serum phosphate levels above 4.5 mg/dL (1.1%). After adjusting for age, creatinine, blood urea nitrogen, uric acid, height, weight, body mass index, systolic and diastolic blood pressure, pulse, cholesterol, alkaline phosphate, aspartate aminotransferase, bilirubin, protein, albumin, hematocrit, and red blood cell distribution width, the fall risk score was more than three times higher in patients with a serum phosphate value below 2.8 mg/dL and was more than three times higher compared to those with a serum phosphate value above 4.5 mg/dL (OR: 3.3, 95%CI:1.5–8.5) (Table 2).

### 3.7. Malnutrition

Pourhassan et al. (2018) [36] conducted a multicenter study on 342 hospitalized patients admitted to a geriatric acute care unit and showed that patients with HP (serum phosphate < 0.68 mmol/L) were significantly more likely to have unintentional weight loss than non-HP patients. They also found that malnutrition or the risk of malnutrition, as assessed by the Mini Nutritional Assessment-Short Form, was present in 86% of HP patients, a value that was significantly higher (*p* = 0.003) than the one observed in non-HP patients (56%). An analysis of the accuracy of the concurrent diagnosis of HP and malnutrition revealed a low sensitivity (9.8%) but a high positive predictive value (86.4%) of HP in relation to malnutrition and a high specificity (97.9%) (Table 2).

### 3.8. Risk of Negative Outcome After Risk of Falls 

Only one of the studies investigated a possible role of HP in the deterioration of the condition of patients after coronary bypass surgery. In a retrospective analysis conducted on a sample of 4989 patients, Park et al. (2019) [33] reported an incidence of 3.7% of all deaths in patients with normal serum phosphate levels and 9.7% in patients with HP (serum phosphate < 2.5 mg/dL); HP was significantly associated with the risk of death in all cases (HR 1.76; 95% CI 1.13–2.76; *p* = 0.01), although this result was not confirmed in the adjusted analysis performed using inverse probability weighting (HR 1.52; 95% CI 0.89–2.61; *p* = 0.12). A higher incidence of graft failure was confirmed in HP patients (Cox regression analysis HR 2.14; 95% CI 1.22–3.75; *p* = 0.01 and IPW analysis HR 2.51; 95% CI 1.37–4.61; *p* = 0.003, respectively) (Table 2).

## 4. Discussion

Electrolyte imbalances are common among older adults and can subtly affect health outcomes, often in ways that are not immediately apparent. Among these electrolytes, phosphorus plays a pivotal role in cellular metabolism, neurological function, and skeletal health. Despite its well-documented physiological functions and the severe effects of hypophosphatemia, there are limited data on the prevalence of non-severe HP in the general population of older adults. This systematic review explores the prevalence and effects of non-severe HP in individuals aged ≥ 65 years without specific diseases causing low phosphate levels.

### 4.1. Frailty

Although the heterogeneity of the included papers, regarding variability in diagnostic criteria for HP and the heterogeneity of the samples examined, may limit the generalizability of the results, our analyses suggest that mild HP can be associated with unhealthy aging, increased mortality, and an increased risk of adverse clinical conditions. In this regard, Fujisawa et al. (2022) [30] highlighted the connection between mild HP and frailty. Frailty is a clinical syndrome that is quite common in older adults, contributing to physical decline, reduced resilience to stressors, and increased risk of adverse outcomes such as falls and hospitalizations [38,39]. According to the findings of Fujisawa et al. [30], HP may be considered a possible marker of frailty and thus may be useful for diagnosing and following up on this condition.

### 4.2. Risk of Falls

Fujisawa et al. also suggested that HP is linked to cognitive impairment, depressed mood, and an increased risk of falls. Both frailty and cognitive decline significantly impact the quality of life and autonomy in older adults [40,41], while falls often lead to severe consequences, including fractures, disability, and even mortality [42,43]. Addressing HP may, therefore, have broader implications for improving the quality and quantity of life in older individuals.

Further evidence supporting the increased risk of falls associated with HP comes from the study by Jang et al. (2022) [32] in hospitalized older adults. Hypophosphatemia may elevate the fall risk by impairing muscle function due to disruptions in ATP production, which is essential for energy in muscle cells [44]. This effect persists even after adjusting for traditional fall risk factors, suggesting that phosphate could serve as a predictive marker for falls.

### 4.3. Infectious or Bacterial Diseases

Infections are a significant concern in older adults, and three of the included studies explored the relationship between HP and infection risk [31,34,37]. Specifically, Morimoto et al. (2022) [31] found that HP was correlated with Legionella spp. infection and severe pneumonia. This may be because HP weakens diaphragm muscle contractility, potentially increasing susceptibility to bacterial pneumonia [45]. Furthermore, low serum phosphate levels may impair host defense mechanisms by reducing phagocytosis and chemotaxis, which can increase the likelihood of secondary infections following influenza [46]. These hypotheses align with Sankaran et al. (1997) [37], who observed that HP was associated with a more severe course of pneumonia, including longer hospitalizations and higher mortality rates.

### 4.4. Malnutrition

The relationship between HP and weight loss, as demonstrated by Pourhassan et al. (2018) [36], suggests that HP may serve as a marker for malnutrition. However, it is important to note that other studies did not explicitly investigate this topic, raising the possibility of bias in the interpretation of their results.

### 4.5. Risk of Negative Outcome After Coronary Artery Bypass Graft

Lastly, the study by Park et al. (2019) [33] indicated that HP is associated with an increased risk of graft failure in patients undergoing coronary artery bypass surgery. This could be due to HP effects on endothelial function and reduced nitric oxide production [47,48], which aligns with previous research suggesting cardiovascular complications in hypophosphatemic patients [49].

### 4.6. Hypophosphatemia Management

In summary, the literature on non-severe HP in older adults is heterogeneous, with varied outcomes and study populations, making direct comparisons challenging. Given this context and the lack of specific recommendations for individuals aged ≥ 65 years without primary causes for HP, we reviewed guidelines for phosphate repletion in adults with HP [50]. In cases of HP, addressing the underlying cause is typically the first step, and this is often sufficient. However, in some instances, phosphate supplementation may be required. When serum phosphorus levels are above 2 mg/dL (0.64 mmol/L), repletion is usually unnecessary unless there are specific indications, such as chronic urinary phosphate loss. For levels below this threshold, treatment is based on symptoms and severity. If phosphorus levels drop to 1–2 mg/dL, asymptomatic patients can be managed with oral phosphate, while symptomatic cases may require intravenous (IV) treatment, transitioning to oral once levels rise above 1.5 mg/dL. For values below 1 mg/dL, IV therapy is initiated, followed by oral repletion when safe levels are achieved.

In clinical practice, the supplementation of non-severe HP, even in asymptomatic patients, is sometimes observed. Given the potential side effects of phosphate supplementation, such as gastrointestinal issues, which can be particularly burdensome for older adults, and the lack of conclusive evidence regarding its benefit in this context, we believe that targeted studies are needed to clarify the prevalence, effects, and therapeutic recommendations for non-severe HP in older individuals.

## 5. Conclusions

### Key Findings and Clinical Implications

This systematic review synthesizes the current literature on non-severe hypophosphatemia (HP) in older adults, highlights its clinical significance, and identifies research gaps. Non-severe HP is associated with frailty, cognitive decline, mood changes, and an increased fall risk. In combination with other electrolyte imbalances, it exacerbates symptoms associated with frailty-related symptoms such as appetite loss, fatigue, muscle weakness, and cognitive impairment. However, its role as a predictive marker for adverse outcomes is only partially understood.

The high prevalence of HP in frail older adults underscores the need for standardized clinical management. Current evidence provides limited guidance on optimal intervention strategies, making it difficult for clinicians to assess the benefits of phosphate repletion. Further research could help refine treatment approaches and improve care for this vulnerable population.

## 6. Future Research Directions

Future research is needed to clarify the prognostic value of serum phosphate levels in assessing frailty and fall risk, particularly in high-risk populations. Prospective studies could help to clarify whether phosphate levels are predictive of health outcomes independently or serve as a marker of underlying disease. In addition, randomized clinical trials could shed light on the potential benefits of phosphate repletion for the functional status and general health of frail older people.

The development of evidence-based guidelines for the treatment of non-severe HP may be worth exploring. Identifying clinically relevant thresholds for phosphate correction and patient subgroups who might benefit most could refine treatment strategies. Investigating the interactions of HP with other electrolyte disorders could further improve the understanding of their role in geriatric care.

Addressing these research gaps may contribute to a more comprehensive understanding of the clinical significance of HP and support future prevention and treatment strategies for aging populations.

## Figures and Tables

**Figure 1 nutrients-17-01354-f001:**
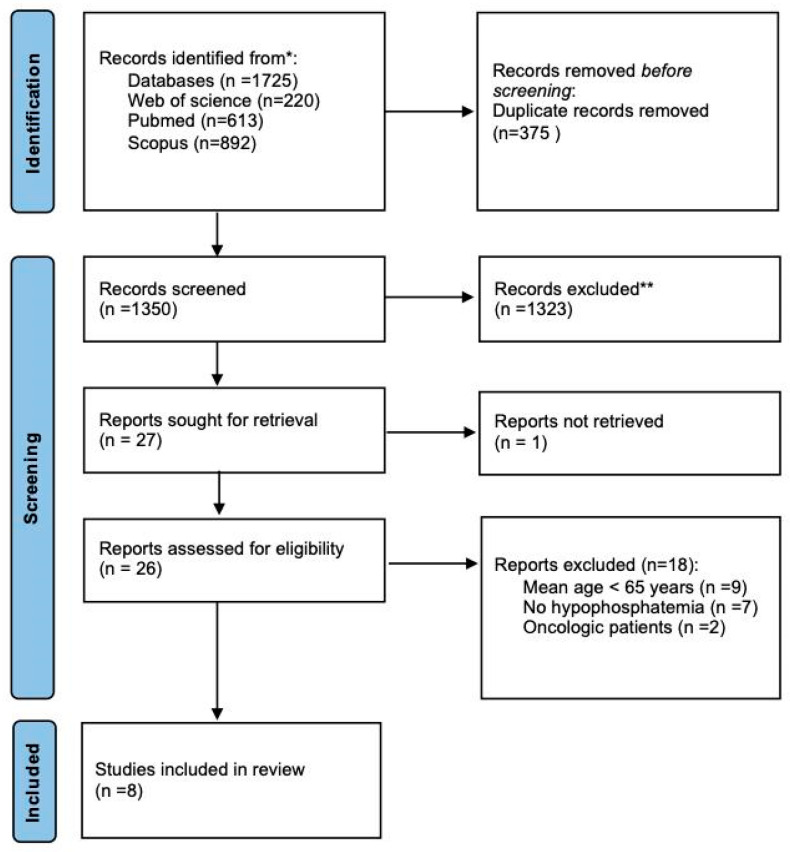
PRISMA flow-chart [29]. * the number of records identified from each database, ** All records were excluded by human.

**Figure 2 nutrients-17-01354-f002:**
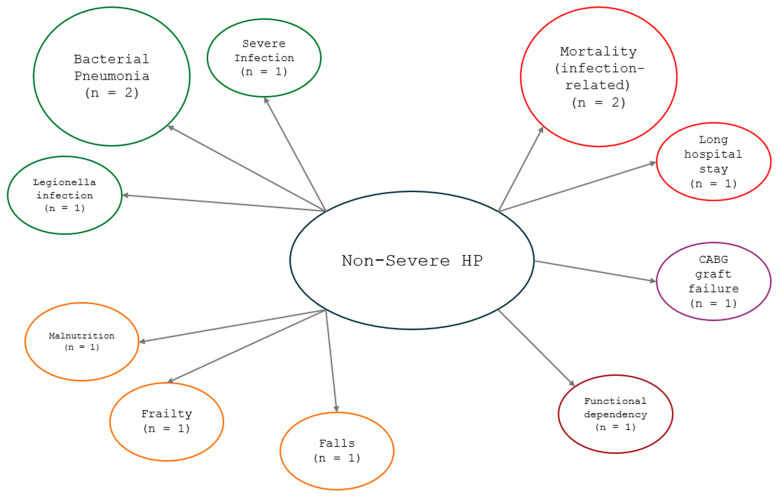
Graphical representation of the main clinical, functional, infectious, surgical, and prognostic outcomes associated with non-severe HP in older adults identified in the included studies. The arrows represent associations reported in the literature. Color coding indicates the type of associated conditions: orange (clinical outcomes), green (infectious conditions), purple (surgical outcomes), brown (functional dependency), and red (prognostic outcomes).

**Figure 3 nutrients-17-01354-f003:**
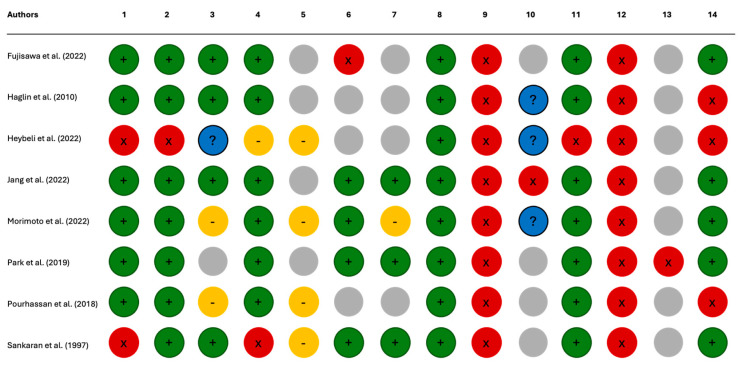
Risk of bias: Fujisawa et al. (2022) [30], Haglin et al. (2010) [34], Heybeli et al. (2022) [35], Jang et al. (2022) [32], Morimoto et al. (2022) [31], Park et al. (2019) [33], Pourhassan et al. (2018) [36], and Sankaran et al. (1997) [37].

**Figure 4 nutrients-17-01354-f004:**
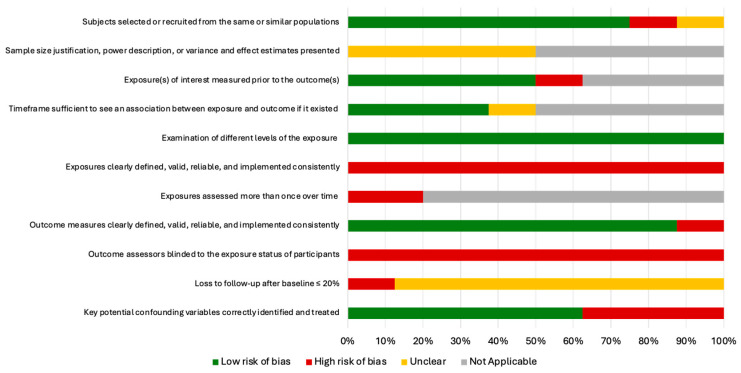
The global quality assessment.

**Table 2 nutrients-17-01354-t002:** Outcome and main findings.

Topic	Authors	Outcome	Main Findings	Conclusion
Hypophosphatemia and related comorbidities	Fujisawa et al. (2022) [30]	Electrolyte imbalance, comorbidities, cognitive function and mood, ADL (both basic and instrumental), physical function, nutrition, fall risks from physical weakness, fall risks from comorbidities	Comparison HP risk between non-frailty group and mildly frail group: OR: 1.52; 95% CI, 1.07–2.14; *p* = 0.02; Comparison HP risk between non-frailty group and moderate frail group: OR: 1.56; 95% CI, 1.08–2.24; *p* = 0.02; Comparison HP risk between non-frail group and severe frail group: OR: 2.00; 95% CI, 1.37–2.89; *p* < 0.001). For every increase of 0.1 in the FI score produced an increase in HP risk of 16% (exp(β) = 1.16, 95%CI: 1.06–1.27). HP correlated with cognitive function and mood (OR: 1.07, 95%CI: 1.03–10.12, *p* < 0.01), fall risks from physical weakness (OR: 1.05, 95%CI: 1.01–1.10, *p* < 0.05), and fall risks from comorbidities (OR: 1.05, 95%CI: 1.00–1.10, *p* < 0.05).	Compared with the non-frail group, the mildly and moderately frail groups tended to have HP.
	Heybeli et al. (2022) [35]	Electrolyte abnormalities in geriatric population with attention to age, comorbidities, drug exposure, and malnutrition	A significantly lower prevalence of diabetes mellitus in patients with HP (13.6%) than in normophosphatemia patients (37%, *p* = 0.026). A greater, but not statistically significant, prevalence of HP in patients over 80 years (6.7%) compared to those in the range 65–79 years (3.7%).	DM was more common in patients with normophosphatemia (*p* = 0.026), while CKD was more present in HP patients (*p* = n.s.).
Infectious or bacterial diseases	Haglin et al. (2010) [34]	Prevalence of HP and role of HP in virus-infected patients in chronic disease and/or bacterial infection and longer hospitalization stay	Prevalence of HP (S-P ≤ 0.82 mmol/L) in 34% of patients with a greater prevalence of mild-severe HP (S-P < 0.70 mmol/L) in 13% of women and 15% of men. Influenza and chronic disease (i.e., DM, asthma bronchiale, CVD, or COPD); double the risk of HP.	A chronic disease in old patients indicates a high prevalence of hypophosphatemia, and this in turn might exacerbate morbidity and mortality.
	Morimoto et al. (2022) [31]	Causative organisms of pneumonia, patient factors, disease severity, and mortality	Non-pulmonary disease and the severity of pneumonia were significantly higher in the patients with HP than in those without it. Polymicrobial infections were lower in HP patients (*n* = 9, 12.5%) than in non-HP patients (*n* = 44, 8.3%). Legionella was more frequent in the HP patients than in the non-HP patients (11.1% vs. 3.6%, *p* = 0.010). Legionella (OR, 2.89; 95% CI, 1.19 to 6.99; *p* = 0.019), diabetes mellitus (OR, 2.53; 95% CI, 1.41 to 4.56; *p* = 0.002), and severe pneumonia (OR, 2.86; 95% CI, 1.57 to 5.22; *p* = 0.001) were independent factors for HP.	HP was not associated with the prognosis in patients with community-acquired pneumonia, although HP could predict abnormal glucose metabolism, Legionella infection, and severe disease.
	Sankaran et al. (1997) [37]	Bacterial pneumonia at discharge, morbidity, and mortality	HP was more prevalent in patients with a diagnosis of bacterial pneumonia at discharge (44.7%) (*p* < 0.001); HP patients had lower levels of potassium, calcium, and albumin compared to their normophosphatemic counterparts, but for different normophosphatemic patients, they had higher glucose levels. Hospitalization was longer for HP patients (HP: 24.6 ± 2.1 days vs. normophosphatemia, 14.1 ± 1.0; *p* < 0.0001), and HP patients had higher mortality rates (*p* < 0.001) than their normophosphatemic counterparts.	HP may be a predictor of the severity of illness in patients admitted to the hospital with bacterial pneumonia.
Risk of falls	Jang et al. (2022) [32]	Risk of falls in hospitalized patients	Fall risk in patients with a lower S-P level (≤2.8 mg/dL) was more than double that of patients with S-P level above 4.5 mg/dL.	Lower s-phosphate level on admission was independently associated with an increased risk of in-hospital falls.
Malnutrition	Pourhassan et al. (2018) [36]	Malnutrition	HP patients had significantly more unintentional weight loss than non-HP patients; malnutrition or at risk of malnutrition was present in 86% of HP patients, values were significantly higher (*p* = 0.003) than reported among patients without HP (56%) among participants.	Older patients with HP are likely to have experienced unintentional weight loss and to have nutritional difficulties compared to non-HP patients.
Risk of negative outcome after CABG	Park et al. (2019) [33]	CABG failure (all-cause death, cardiovascular death, graft failure, composite of major adverse cardiovascular and cerebral events MACE)	A 3.7% incidence of all-cause death in patients with normal S-P levels, and 9.7% in patients with HP (S-P lower than 2.5 mg/dL); the HP was significantly associated with risk of all-cause (HR 1.76; 95% CI 1.13–2.76; *p* = 0.01) in patients with HP; a higher incidence of graft failure was confirmed hypophosphatemia patients (multivariate Cox regression HR 2.14; 95% CI 1.22–3.75; *p* = 0.01, IWR analysis HR 2.51; 95% CI 1.37–4.61; *p* = 0.003).	Preoperative serum phosphorus abnormalities were not associated with outcomes after CABG except for graft failure.

## Data Availability

All the data are available to the public and presented in the manuscript.

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
