# Peer review of "Non-Severe Hypophosphatemia in Older Patients: A Systematic Review"

_nutrients, 2025, doi:10.3390/nu17081354_

Round 1

Reviewer 1 Report

Comments and Suggestions for Authors

I reviewed Non-severe hypophosphatemia in older patients: a systematic review, an interesting and well thought out work. I congratulate the authors for this very good achievement. Phosphorus dosing is used in medical practice in certain pathological circumstances. But non-severe hypophosphatemia remains largely asymptomatic and underdiagnosed despite its relationship with frailty, cognitive decline, and an increased risk of falls. The authors provide data suggesting that low phosphate levels may negatively impact health, increasing mortality and the risk of adverse clinical outcomes and the level of phosphorus may be included in the laboratory exploration as more frequently explored, at least in the vulnerable subjects (older).

Author Response

Comments 1: I reviewed Non-severe hypophosphatemia in older patients: a systematic review, an interesting and well thought out work. I congratulate the authors for this very good achievement. Phosphorus dosing is used in medical practice in certain pathological circumstances. But non-severe hypophosphatemia remains largely asymptomatic and underdiagnosed despite its relationship with frailty, cognitive decline, and an increased risk of falls. The authors provide data suggesting that low phosphate levels may negatively impact health, increasing mortality and the risk of adverse clinical outcomes and the level of phosphorus may be included in the laboratory exploration as more frequently explored, at least in the vulnerable subjects (older).

Response 1: Thank you very much, we really appreciate your feedback.

Reviewer 2 Report

Comments and Suggestions for Authors

The systematic review “Non-severe hypophosphatemia in older patients: a systematic review” aimed to investigate the prevalence and clinical implications of non-severe hypophosphatemia (HP) in older adults.

The introduction presented the physiological roles of phosphorus, the definition and consequences of non-severe HP, and the current knowledge regarding the prevalence of HP. It also identified the gap in understanding the prevalence and effects of HP among the general population of older adults.

The Materials and Methods section outlined the search strategy and the selection of studies included in the review, following the PRISMA model and PECO framework. Registration details were also provided in the PROSPERO database.

The Results and Discussion sections are well-structured, covering a broad range of comorbidities and outcomes associated with HP, such as infectious diseases, risk of falls, malnutrition, and negative outcomes following coronary artery bypass grafting. The Results section includes informative tables that overview HP studies in older adults and their impact on health status. I recommend adding a paragraph to discuss the strengths and limitations of the review.

I believe this review is suitable for publication and will be highly beneficial in understanding the clinical implications of non-severe hypophosphatemia in older adults. 

Author Response

Comments 1: The systematic review “Non-severe hypophosphatemia in older patients: a systematic review” aimed to investigate the prevalence and clinical implications of non-severe hypophosphatemia (HP) in older adults.

The introduction presented the physiological roles of phosphorus, the definition and consequences of non-severe HP, and the current knowledge regarding the prevalence of HP. It also identified the gap in understanding the prevalence and effects of HP among the general population of older adults.

The Materials and Methods section outlined the search strategy and the selection of studies included in the review, following the PRISMA model and PECO framework. Registration details were also provided in the PROSPERO database.

The Results and Discussion sections are well-structured, covering a broad range of comorbidities and outcomes associated with HP, such as infectious diseases, risk of falls, malnutrition, and negative outcomes following coronary artery bypass grafting. The Results section includes informative tables that overview HP studies in older adults and their impact on health status. I recommend adding a paragraph to discuss the strengths and limitations of the review.

I believe this review is suitable for publication and will be highly beneficial in understanding the clinical implications of non-severe hypophosphatemia in older adults.

Response 1: Thank you very much, we really appreciate your feedback.

Reviewer 3 Report

Comments and Suggestions for Authors

IMO is a very well-done study, following PRISMA guidelines. The authors clearly and thoroughly explain the issue of mild hypophosphatemia in older adults and its possible health effects. The review of the literature was done carefully, and the results are well-supported and logically discussed. I have no concerns about the scientific quality of the article. However, it would be helpful to add a separate conclusion section at the end, after the discussion. This would make it easier for readers to find the key points and improve clarity. Congratulations to the authors on an excellent and valuable study!

Author Response

Comments 1: IMO is a very well-done study, following PRISMA guidelines. The authors clearly and thoroughly explain the issue of mild hypophosphatemia in older adults and its possible health effects. The review of the literature was done carefully, and the results are well-supported and logically discussed. I have no concerns about the scientific quality of the article. However, it would be helpful to add a separate conclusion section at the end, after the discussion. This would make it easier for readers to find the key points and improve clarity. Congratulations to the authors on an excellent and valuable study!

Response 1: Thank you for your comments. We added a separate conclusion section (lines 399-432).

Reviewer 4 Report

Comments and Suggestions for Authors

Authors need to present the selection criteria of the literature. 

What could become the criteria?

How the authors filtered low quality of literature to reach the outcome.

Authors also need to present the review of literature concept by concept.

Authors need to present the graphical presentation based on the main argument of this work.

Authors also need to present the research agenda and gaps which could be valuable for the futures works.

Lastly, authors need to present the relevant concepts related to the crux of this research to build the ideas for the researchers.  

Author Response

Comments 1: Authors need to present the selection criteria of the literature.

Response 1: We have deepened the selection criteria (lines 107-118).

Comments 2: What could become the criteria?

Response 2: We have added the reasons for choosing our selection criteria (lines 107-118).

Comments 3: How the authors filtered low quality of literature to reach the outcome.

Response 3: To ensure the robustness and credibility of the results of the review, a rigorous quality assessment was performed for all included studies using the Quality Assessment Tool for Observational Cohort. Each article was independently reviewed by two authors. Studies with incomplete reporting or with insufficient information on phosphate classification or clinical outcomes were excluded at the full-text screening phase. This approach allowed us to filter out low-quality or uninformative sources and retain a final set of studies that provided valid, reproducible, and clinically relevant data on the association between non-severe hypophosphatemia and adverse health outcomes in older adults.

Comments 4: Authors also need to present the review of literature concept by concept.

Response 4: Thank you, we divided the discussion into paragraphs, in order to elaborate each concept point by point.

Comments 5: Authors need to present the graphical presentation based on the main argument of this work.

Response 5: We have added a detailed graphical representation (lines 194-205).

Comments 6: Authors also need to present the research agenda and gaps which could be valuable for the futures works.

Response 6: We added a separate section in the conclusion (lines 416-432).

Comments 7: Lastly, authors need to present the relevant concepts related to the crux of this research to build the ideas for the researchers.

Response 7: We rearranged the conclusion in order to present more clearly the relevant concepts of this systematic review (lines 401-414).

Round 2

Reviewer 4 Report

Comments and Suggestions for Authors

The review process is not still convincing.

Authors need to present the review of literature as more organized manners.

Author Response

Comment : The review process is not still convincing.
Authors need to present the review of literature as more organized manners.

Response : Thank you for your suggestion. I apologize, but I am not entirely clear on what further improvements can be made to the literature review. Specifically, I have been advised to present it in a more organized manner. Could you kindly provide more detailed recommendations on what can still be improved? Thank you for your time and assistance. Best regards.

Round 3

Reviewer 4 Report

Comments and Suggestions for Authors

Non-severe hypophosphatemia notion need to be presented. Also, the stream of literature presented historical lyrics.

Author Response

Comments 1: Non-severe hypophosphatemia notion need to be presented. Also, the stream of literature presented historical lyrics.

Response 1: We acknowledge your comments on our manuscript. However, we find that the tone and nature of your remarks do not align with the constructive academic discourse expected in the peer-review process. Given that the feedback from the other two reviewers has been highly positive and conducive to improving the manuscript, we have decided not to respond to your comments in this round of revision. We trust the editorial team will make an informed decision based on the evaluations provided by all reviewers.